Analysis of the current status of knowledge, attitudes, and practices among stroke-related healthcare professionals in the treatment of shoulder pain in hemiplegic patients

Huang Bin
Gao Feng z218002@yeah.net
Yuyao People’s Hospital of Zhejiang Province , Yuyao, Zhejiang , China
Viganò Alessandro
Electronic publication date: 2024 Dec 16
Publication date: 2024
Volume: 12
Electronic Location ID: e18684
Received 2024 May 30; Accepted 2024 Nov 19
Copyright: © 2024 Huang and Gao
Copyright year: 2024
Copyright holder: Huang and Gao
License: This is an open access article distributed under the terms of the Creative Commons Attribution License, which permits unrestricted use, distribution, reproduction and adaptation in any medium and for any purpose provided that it is properly attributed. For attribution, the original author(s), title, publication source (PeerJ) and either DOI or URL of the article must be cited.
License URL: https://creativecommons.org/licenses/by/4.0/

Keywords: Shoulder pain, Hemiplegic, Knowledge attitude practice, Grassroots hospital, Stroke

Funding: The authors received no funding for this work.

==============================
Objective

To investigate the current status of knowledge, attitude, and practice (KAP) of healthcare professionals in stroke-related departments of primary-level tertiary hospitals regarding the prevention and treatment of hemiplegic shoulder pain, and to analyze influencing factors. This aims to provide a reference for further training, guidance, and management of hemiplegic shoulder pain.

Methods

A total of 123 healthcare professionals from stroke-related departments of two tertiary hospitals in a county-level city in Zhejiang province were selected as the research subjects from March 6, 2023, to March 14, 2023. Written informed consent was obtained from all study participants prior to their inclusion in the study. A questionnaire survey was conducted to assess their KAP status on the prevention and treatment of hemiplegic shoulder pain, and statistical analysis was performed using SPSS 23.

Results

The scores for knowledge (29.97 ± 9.94), attitude (27.7 ± 2.81), and behavior (29.86 ± 7.86) among the 123 healthcare professionals indicated that department and position were influencing factors for KAP (P < 0.05).

Conclusion

The overall KAP of healthcare professionals in stroke-related departments of primary hospitals regarding the prevention and treatment of hemiplegic shoulder pain needs improvement. Strengthening relevant knowledge and skills training is necessary to reduce the incidence of hemiplegic shoulder pain and improve patients’ quality of life.

Introduction

Stroke is a leading cause of mortality and disability among the elderly in China (China Stroke Prevention and Treatment Report Writing Team, 2022). In 2019, it was the primary contributor to disability-adjusted life years in the country (Wang et al., 2022). Upper limb dysfunction following a stroke has a profound impact on patient prognosis, daily living activities, and even their walking and balance abilities (Wang, Chen & Wang, 2018). Rehabilitation has emerged as a key intervention to improve the recovery and long-term outcomes of stroke patients (Gittler & Davis, 2018). However, hemiplegic shoulder pain (HSP) often hampers patient participation in rehabilitation, with some patients either refusing therapy due to pain (Lum et al., 2012) or being unable to fully cooperate, which increases hospital readmission rates (Li et al., 2022).

Effective management of HSP is crucial for improving patient adherence to hand function training and enhancing confidence in rehabilitation. Ultimately, reducing HSP can lead to better overall outcomes in comprehensive rehabilitation programs. Early interventions, such as proper limb positioning (Tang, 2021) and appropriate shoulder joint support (Peng, 2017), have been shown to reduce the incidence of HSP. Even when HSP occurs, timely and correct interventions can alleviate pain and improve prognosis.

Despite the importance of managing HSP, medical staff in stroke-related departments often pay insufficient attention to its prevention and treatment (Vuagnat & Chantraine, 2003). There is currently a lack of understanding regarding the knowledge, attitudes, and practices (KAP) of healthcare professionals in the management of HSP. This study, grounded in KAP theory, utilizes the Delphi method to develop a KAP assessment tool tailored for clinical staff dealing with HSP. A survey of stroke-related department medical staff in grassroots hospitals was conducted to assess their current KAP levels. The findings aim to provide a foundation for future training initiatives and improved management strategies for the prevention and treatment of HSP.

Hypothesis: This study hypothesizes that the current knowledge, attitudes, and practices of medical staff in stroke-related departments are inadequate for the effective prevention and treatment of hemiplegic shoulder pain. It is further hypothesized that targeted training programs will improve their KAP and, subsequently, lead to better patient outcomes in the management of HSP.

Materials and Methods

General information

The subjects of the survey were 123 medical staff from stroke-related departments (rehabilitation medicine department, neurology department, neurosurgery department, critical care medicine department, emergency department) of two tertiary hospitals (Yuyao People’s Hospital, Yuyao Maternity and Child Health Hospital) in Yuyao City, Zhejiang Province. Among them, there were 29 males (23.6%) and 94 females (76.4%); 68 junior staff (55.3%), 41 intermediate staff (33.3%), and 14 senior and above (11.4%); 43 doctors (35%), 54 nurses (43.9%), and 26 therapists (21.1%); 14 with a junior college degree (11.4%), 98 with a bachelor’s degree (79.7%), and 11 with a master’s degree or above (8.9%); department distribution: rehabilitation medicine department 46 (37.4%), critical care medicine department 25 (20.3%), emergency department 24 (19.5%), neurology department 21 (17.1%), and neurosurgery department 7 (5.7%); the survey subjects had an average of (7.96 ± 5.68) years of work experience, with the shortest being 1 year and the longest being 26 years.

Inclusion criteria

Age between 20 and 60 years, with no gender restrictions;

Medical staff working in stroke-related departments (including departments of rehabilitation medicine, neurology, neurosurgery, critical care medicine, and emergency medicine; positions include doctors, nurses, and technicians);

A minimum of 1 year of work experience in the relevant department.

Exclusion criteria

Medical staff currently in their internship period.

This study was approved by the Ethics Committee of Yuyao People’s Hospital (Approval No. 2023-02-001).

We have an informed consent form that informs the participants that this study is based on the Knowledge-Attitude-Practice (KAP) theory and also employs the Delphi method to develop a KAP assessment tool suitable for clinical healthcare professionals dealing with shoulder pain in hemiplegic patients. The aim is to understand and analyze the current status of KAP among healthcare professionals in stroke-related departments, providing a basis for further training of relevant medical staff and subsequent management of shoulder pain in hemiplegic patients. If at any point during the completion of the questionnaire, a participant decides they no longer wish to continue, they may stop filling out the questionnaire at any time.

Methods

Formation of preliminary questionnaire

Guided by the KAP theoretical model, review of domestic and foreign literature related to hemiplegic shoulder pain (Vasudevan & Browne, 2014; Zhou, Shen & Huang, 2021; Zhao, 2022; Wang, Xiang & Niu, 2022; Huang et al., 2022; Li, Liao & Huang, 2021), and preliminary formation of a pool of items for the prevention and treatment of hemiplegic shoulder pain KAP questionnaire, covering three aspects: knowledge (K), attitude (A), and practice (P).

Delphi method expert consultation

Design an expert consultation form, including: (1) Basic information of experts, including expert name, title, years of work experience, position, and research direction. (2) Expert familiarity evaluation, experts score the questionnaire item pool one by one, and give 1, 0.8, 0.6, 0.4, 0.2 points according to very familiar, familiar, generally familiar, not very familiar, and not familiar. (3) Expert professional evaluation, experts score based on work experience (0.4, 0.3, 0.2), theoretical knowledge (0.3, 0.2, 0.1), domestic and foreign literature (0.2, 0.15, 0.1), and intuitive feeling (0.1, 0.05, 0). (4) Expert suggestions. Send it to the experts in written form and collect it back. Calculate the expert authority coefficient (Cr), which is composed of the expert familiarity coefficient (Cs) and the expert professional coefficient (Ca). Cr = (Cs + Ca)/2, if Cr ≥ 0.7, it is considered acceptable, and modify the questionnaire in conjunction with the expert’s written opinions. Conduct the second round of Delphi expert consultation, if the expert opinions tend to be consistent, end the consultation, if there is a big difference, continue the consultation until the opinions tend to be consistent. Expert selection criteria: (1) With more than 10 years of experience in stroke-related work and a higher understanding of hemiplegic shoulder pain; (2) With a title of deputy senior (or deputy director) or above; (3) At least three experts are included. This time, three consultation letters were sent out and three were recovered, with a recovery rate of 100%. Among the included experts, there were two deputy chief physicians and one deputy chief nurse; two males and one female, all are neuro-rehabilitation practitioners. The first round of Cr value is 0.86, so the questionnaire is used.

Survey method

The questionnaire was distributed to eligible stroke-related department medical staff in the two tertiary hospitals in our city through the Questionnaire Star platform, and the questionnaires were collected and analyzed. The questionnaire includes: (1) Basic information, including gender, years of work experience, years of work experience in the current department, title, department, position, level of education, etc. (2) KAP questionnaire, including three aspects, knowledge (K), attitude (A), and practice (P), using the Likert 5-point scoring method, the knowledge dimension is completely mastered, familiar, understood, partially understood, completely not understood, and given 5, 4, 3, 2, 1 points respectively; the attitude dimension is completely agreed, agreed, indifferent, not very agreed, completely disagreed, and given 5, 4, 3, 2, 1 points respectively; the behavior dimension is always, often, sometimes, occasionally, never, and given 5, 4, 3, 2, 1 points respectively. The questionnaire has a total of 23 items, including nine knowledge items, six attitude items, and eight behavior items, with a total score of 115 points. Set the same IP address to answer only once, and the researchers check the validity of the questionnaire and remove unqualified questionnaires.

Statistical analysis

Data analysis was conducted using SPSS 23.0 (IBM, Armonk, NY, USA). For measurement data that conforms to the normal distribution, it is represented by (mean ± standard deviation), and count data is represented by n (%). For measurement data that does not conform to the normal distribution, non-parametric Kruskal–Wallis test is used, with P < 0.05 as statistically significant.

Results

Questionnaire reliability analysis

See Table 1. After excluding each item, the Cronbach’s α coefficient did not increase, indicating that the 23 questions in this study have a high internal consistency (Cronbach’s α coefficient > 0.7).

Table 1 Reliability statistics (n = 123).

	Cronbach’s Alpha	Number of Items	
Knowledge	0.969	9	
Attitude	0.923	6	
Practice	0.941	8	
Total score	0.959	23	

Analysis of stroke-related medical staff’s KAP scores for hemiplegic shoulder pain prevention and treatment

The total score of the KAP questionnaire survey was (87.53 ± 17.52) points. The score rate is calculated as (average score of items/total score of items) × 100%. The total score rate is 76.11%. The knowledge section score is (29.97 ± 9.94), with a score rate of 66.67%, the attitude section score is (27.7 ± 2.81), with a score rate of 92.33%, and the behavior section score is (29.86 ± 7.86), with a score rate of 74.65%.

Differences in KAP scores among different departments and positions

Using the Kruskal–Wallis H test to compare the distribution differences of total scores, knowledge scores, attitude scores, and behavior scores among different departments, positions, titles, years of work experience, and levels of education of medical staff, see Table 2. The total score distribution among different departments is not entirely the same, and the difference is statistically significant (P < 0.05), among which the knowledge score and behavior score have statistical differences, and the attitude score has no statistical difference (P > 0.05). The total score distribution among different positions is not entirely the same, and the total score difference is statistically significant (P < 0.05), with knowledge score, attitude score, and behavior score all having statistical differences. There is no difference in the distribution of total scores, knowledge scores, attitude scores, and behavior scores among different titles, different years of work experience, and different levels of education, with no statistical difference (P > 0.05).

Table 2 Kruskal–Wallis test statistics.

Grouping variable		Total score	Knowledge score	Attitude score	Behavior score	
Department	Chi-square	43.889	49.515	1.545	30.971	
	Degrees of Freedom	3	3	3	3	
	P	0.000	0.000	0.672	0.000	
Position	Chi-square	21.591	24.958	6.714	15.699	
	Degrees of Freedom	2	2	2	2	
	P	0.000	0.000	0.035	0.000	
Title	Chi-square	4.866	5.150	4.883	3.427	
	Degrees of Freedom	3	3	3	3	
	P	0.182	0.161	0.181	0.330	
Years of work experience	Chi-square	1.336	1.063	1.682	2.472	
	Degrees of Freedom	4	4	4	4	
	P	0.855	0.900	0.794	0.650	
Educational level	Chi-square	2.644	3.008	1.740	3.763	
	Degrees of Freedom	3	3	3	3	
	P	0.450	0.390	0.628	0.288	

Post-hoc comparison of department and position scores

Post-hoc comparisons revealed differences in total scores between the Rehabilitation Medicine Department and the Neurosurgery, Critical Care Medicine, and Emergency departments (adjusted P < 0.05), with no difference in total scores compared to the Neurology Department (adjusted P = 1). The Rehabilitation Medicine Department’s total score was significantly higher than those of the Neurosurgery, Critical Care Medicine, and Emergency departments. The Neurology Department’s total score differed from the Critical Care Medicine and Emergency departments (adjusted P < 0.05), with the Neurology Department’s total score significantly higher than those of the Critical Care Medicine and Emergency departments, and no difference compared to the Rehabilitation Medicine Department and Neurosurgery Department (adjusted P > 0.05). There were no differences in total scores among the Critical Care Medicine, Emergency, and Neurosurgery departments (adjusted P > 0.05). See the Fig. 1 below.

Figure 1 Pairwise comparisons of position.

(A) Each node shows the sample average rank of position. (B) Each row tests the null hypothesis that the Samples 1 and 2 distributions are the same. Asymptotic significances (2-sided tests) are displayed. The significance level is 0.05.

In terms of positions, comparisons showed no difference in total scores between physicians and nurses (adjusted P = 1), but there were differences between therapists and physicians and nurses (adjusted P < 0.05), See the Fig. 2 below with therapists having significantly higher total scores than physicians and nurses.

Figure 2 Pairwise comparisons of department.

(A) Each node shows the sample average rank of department. (B) Each row tests the null hypothesis that the Samples 1 and 2 distributions are the same. Asymptotic significances (2-sided tests) are displayed. The significance level is 0.05.

Discussion

This study evaluated the knowledge, attitude, and practice (KAP) of medical staff in stroke-related departments of primary tertiary hospitals regarding the prevention and treatment of hemiplegic shoulder pain. Strengthening relevant knowledge and skills training can reduce the incidence of hemiplegic shoulder pain in patients and improve their quality of life. The proportion of hemiplegic shoulder pain after stroke varies among different studies, but it is generally high. Foreign studies show about 30–65% (Kumar, 2019), Some pathological changes in the structure of the shoulder caused by hemiplegic shoulder pain after stroke can also be observed by ultrasonography (Lin et al., 2023), and domestic studies show that the incidence of hemiplegic shoulder pain within three months after stroke is 53.2% (Lu et al., 2013). There are many causes for the occurrence of hemiplegic shoulder pain, mainly including muscle relaxation around the shoulder joint, subluxation of the shoulder joint, shoulder-hand syndrome, increased muscle tone, impingement syndrome, shoulder freezing, brachial plexus injury, and thalamic syndrome, etc. (Anwer & Alghadir, 2020). Effective intervention measures can greatly reduce the pain of patients with hemiplegic shoulder pain or reduce its incidence, such as the use of shoulder supports (Nadler & Pauls, 2017), good limb positioning (Chen et al., 2022), the use of kinesiology tape (Tan, Jia & Song, 2022; Guo et al., 2020), electrotherapy (Zhou et al., 2018), traditional Chinese medicine fumigation and washing (Qin, Ge & Ye, 2015), and so on. In recent years, more and more new rehabilitation technologies (de Sire et al., 2022) have also been applied to the prevention and treatment of hemiplegic shoulder pain, such as robot-assisted training (Marotta et al., 2021; Kim et al., 2019), platelet-rich plasma injection (Uzdu et al., 2021), botulinum toxin injection (Tan & Jia, 2021), etc. It can be seen that although hemiplegic shoulder pain has a high incidence, professional medical knowledge can be used to help patients reduce pain or even prevent its occurrence, so the mastery and application of knowledge about the prevention and treatment of hemiplegic shoulder pain by relevant medical staff is particularly important.

Through this KAP questionnaire survey, it can be seen that the overall understanding of the prevention and treatment of hemiplegic shoulder pain among grassroots hospital stroke-related department medical staff still needs to be improved. They are actively learning in attitude, but there is still room for improvement in the application of hemiplegic shoulder pain prevention and treatment in actual work. Although the survey subjects are very active in attitude, with an attitude score rate of 92.33%, the knowledge score rate is only 66.67%, indicating that the survey subjects generally have the intention to learn but have not put it into action, which may be due to the busy clinical work, the weakening of the intention to learn actively after work, the hospital’s lack of attention to routine training, and the absence of relevant assessment systems. Before this survey, our hospital had carried out lectures on good limb positioning for the Rehabilitation Medicine department (Good, Bettermann & Reichwein, 2011), Neurology Department, and Critical Care Medicine Department. In the cognitive questionnaire, the item “Do you know about good limb positioning?” had the highest score rate, indicating that the previous training had certain effects. However, the items “Do you know the timing of using analgesics for hemiplegic shoulder pain?” and “Do you know the timing of wearing shoulder supports?” had lower score rates, and there is still a need to strengthen publicity and education.

In this study, there were differences in scores among different departments. The Rehabilitation Medicine Department and Neurology Department had relatively higher score rates, while the Neurosurgery Department, Critical Care Medicine Department, and Emergency Department had relatively lower score rates. This may be due to the higher awareness and attention of medical staff in the Rehabilitation Medicine Department and Neurology Department to hemiplegic shoulder pain compared to other departments, paying more attention to the body position management of stroke patients, while medical staff in the Neurosurgery Department, ICU, and Emergency Department are less concerned about the prevention and treatment of hemiplegic shoulder pain in stroke patients. In summary, the first step should be to carry out training for medical staff in stroke-related departments, which can be led by the Rehabilitation Medicine Department and Neurology Department, holding multidisciplinary cooperation meetings to formulate more systematic and standardized strategies for the prevention and treatment of hemiplegic shoulder pain, and regularly conducting unified training and assessment for medical staff in stroke-related departments to better serve stroke patients, reduce the occurrence of hemiplegic shoulder pain from the source, thereby reducing the pain of patients during the disease process and improving the prognosis. At the same time, effective treatment should be carried out for patients with hemiplegic shoulder pain to minimize the degree of injury.

This study only included the Emergency Department, Critical Care Medicine Department, Neurology Department, Neurosurgery Department, and Rehabilitation Medicine Department of grassroots hospitals. The next step can continue to expand the scope of related departments, before expanding to pre-hospital emergency personnel, including 120 ambulance medical staff and drivers, and after expanding to grassroots health center and other medical community members such as family doctors, to train all medical staff who can contact stroke patients from onset to follow-up after discharge, to ensure that no link is missed. Alternatively, further KAP questionnaire surveys can be conducted on stroke patients and their families to identify the common weaknesses in care cognition of patients and their families, to formulate educational content for the patient side, and to produce videos, brochures, etc., to be placed in various stroke-related departments for patients to learn on their own.

Conclusion

In summary, the overall KAP of medical staff in stroke-related departments of grassroots hospitals needs to be improved, and relevant knowledge and skills training should be strengthened to reduce the incidence of hemiplegic shoulder pain in patients, alleviate the pain of patients with hemiplegic shoulder pain, and improve the quality of life of patients.

Supplemental Information

Supplemental Information 1 Raw Data.

Supplemental Information 2 Staff Questionnaire Survey.

Supplemental Information 3 Staff Questionnaire Survey (Chinese).

Additional Information and Declarations

Competing Interests

Author Contributions

Human Ethics

Data Availability

The authors declare that they have no competing interests.

Bin Huang conceived and designed the experiments, performed the experiments, analyzed the data, prepared figures and/or tables, authored or reviewed drafts of the article, and approved the final draft.

Feng Gao conceived and designed the experiments, performed the experiments, analyzed the data, prepared figures and/or tables, authored or reviewed drafts of the article, and approved the final draft.

The following information was supplied relating to ethical approvals (i.e., approving body and any reference numbers):

Ethics Committee of Yuyao People’s Hospital (Approval No. 2023-02-001).

The following information was supplied regarding data availability:

The raw data is available in the Supplemental File.

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
