# Peer review of "Analysis of the current status of knowledge, attitudes, and practices among stroke-related healthcare professionals in the treatment of shoulder pain in hemiplegic patients"

_PeerJ, doi:10.7717/peerj.18684_

## Round 0.1 · original submission · Major Revisions

Based on reviewers’ suggestions, I would recommend to revise the manuscript especially in the light of stressing the rational of the study and its study design and methods.

Reviewer 1 ·

Basic reporting

This is an interesting article. I have several comments to add.

First, regarding the prevalence of different shoulder pathologies in patients with hemiplegic shoulder pain, the authors should mention more about the use of ultrasound imaging on stroke patients.

Second, the authors should discuss more about the management of different injection techniques for the treatment of hemiplegic shoulder pain:

Third, the hypothesis of the present study should be given at the end of the introduction.

Fourth, the general information of the participants should be described in the results section.

Fifth, the inclusion and exclusion criteria need to be presented in complete sentences.

Experimental design

.

Validity of the findings

.

·

Basic reporting

The article meets professional standards, with a structured layout having figures and tables that illustrate essential findings. in addition, the availability of source data enhances openness of the research work hence making it possible for anybody willing to do further analysis. The study findings are relevant as they have addressed its objectives while at the same time indicating some grey zones that need be looked at by relating professionals in medicine so as make them better than they are currently.

In my view, this paper offers a useful contribution to available literature within this area and provides the foundation for future research and translation into clinical practice. The systematic approach to sampling health professionals' views and practices contributes knowledge to the issue and raises connected issues related to the importance of appropriate interventions in the management of shoulder pain in hemiplegic stroke patients. Future research could reasonably extend such findings to give greater emphasis on the development of targeted education for its application in rehabilitation settings in order to achieve the best possible results for patients.

Experimental design

No comments

Validity of the findings

No comments

Additional comments

The authors discuss with great care healthcare professionals' attitudes and opinions on the management of hemiplegic shoulder pain, thus providing an overview of available knowledge, attitudes, and current clinical practice.

In view of the professional standards of writing for an academic context, this article is crystal clear and systematic in its reporting. The introduction offers some important background information to place the study into a broader perspective of research on rehabilitation after stroke, thus making it easier to understand the topic's significance. This is a methodologically quite sound design, with surveys as a tool for data collection from an appropriate sample of healthcare professionals. The results are logically reported with proper figures and tables contributing to the better understanding of the findings.

The discussion section of the article goes on to take results and again critically analyses them in light of existing literature, pointing out gaps and the redundancy or implications for further research and practice. Conclusions are therefore very much data-based, giving valuable insight into the challenges and opportunities in the management of shoulder pain in hemiplegic stroke patients.

This is an important paper that contributes to the literature and can enable actionable insights for the improvement of clinical practices for the care of persons with stroke.

·

Basic reporting

The study provides a clear background on the importance of managing shoulder pain in hemiplegic stroke patients, highlighting the impact on rehabilitation outcomes. However, the introduction needs restructuring, and the relevant literature is not cited appropriately. The text also exhibits major issues with flow and coherence that would benefit from additional editing for clarity.
Comments:
• The overall study requires reorganization. The English writing necessitates proofreading and editing.
• To improve the clarity and flow of the introduction, the sentence structures should be revised, and the introduction should be structured in paragraphs.
• Providing a critical review of similar existing studies is important to address the knowledge gap in the current understanding.
• The objective of the study should be stated clearly and explicitly.

Experimental design

Comments:
• The study design is unclear.
• Consider providing more detailed information on the inclusion and exclusion criteria for participant selection to enhance the reproducibility of the study.
• The authors should clarify whether the study's sample size was determined through a power analysis, as this would support the adequacy of the sample size used.
• Important ethical considerations, such as, obtaining informed consent from participants were not provided!
• Guidelines for reporting the study was not followed.
• The methods used for selecting participants is unclear.

Validity of the findings

Comments:
• Participants demographic characteristics were not provided.
• The authors should discuss the potential limitations of the study in greater detail, particularly regarding the generalizability of the findings to different healthcare settings.
• It would be beneficial to include a discussion on the potential impact of confounding variables and how they were controlled for in the analysis.

Additional comments

The study provides valuable insights into the management of shoulder pain in hemiplegic stroke patients, a crucial topic for improving patient outcomes. The findings have practical implications for developing targeted training programs for healthcare professionals. However, the manuscript would benefit from addressing the previously provided comments.

·

Basic reporting

lines 82-83: the author said that medical staff do not pay enough attention to the prevention and treatment of hemiplegic shoulder pain, is there any data about that?

Experimental design

using the KAP questionnaire, how did the author consider the gaps between what the respondent said and what was done?

Validity of the findings

no comment

Additional comments

no comment

Reviewer 5 ·

Basic reporting

Please refer to "4. Additional comments"

Experimental design

Please refer to "4. Additional comments"

Validity of the findings

Please refer to "4. Additional comments"

Additional comments

This study aims to investigate the professionals’ knowledge, attitude, and practice (KAP) in stroke-related departments regarding the treatments of hemiplegic shoulder pain. To test it, 123 healthcare professionals are recruited, and a questionnaire is used to evaluate the KAP. The results indicate that department and position influence the KAP. It is important to enhance the overall KAP of healthcare professionals in stroke-related departments. While potentially interesting, this work suffers from several major problems.
1. Without the clear rationale/hypothesis, the study potentially leads to a lack of logic and coherence.
2. What is the significance of this study?
3. There is no explanation provided for the determination of sample sizes.
4. Some references are not in English, making it difficult to evaluate the conclusions drawn by the authors.
5. The authors should provide more details regarding the participant and expert inclusion/exclusion criteria.
6. Substantial information is missing in the section of Materials and Methods. For example, how to calculate KAP scores? Detailed statistical analyses are required for each test.
7. In the Conclusion, the authors mention that “…relevant knowledge and skills training should be strengthened to reduce the incidence of hemiplegic shoulder pain in patients, alleviate the pain of patients with hemiplegic shoulder pain, and improve the quality of life of patients”. While testing correlations is interesting, it is more crucial to examine causality. Additionally, correlation is not causality. Hence, the current findings do not support the conclusions.
8. Figures are too blurry, please update with clear ones.
9. This manuscript requires proofreading.

---

## Round 0.2 · Minor Revisions

Please, consider the suggestions by reviewer 1 to be implemented in the paper. Consider also that the suggested citation is not strictly referred to the paper’s subject and so I find it potentially appropriate but not mandatory to include it. However, being an orthopedic matter and not neurological, I am not expert to deny it's validity. So I recommend to add it the paper, unless you have a clear objection.

Reviewer 1 ·

Basic reporting

See below

Experimental design

See below

Validity of the findings

See below

Additional comments

The article is compelling, and I would recommend its publication if the authors address the following suggestions:

Many patients with hemiplegic shoulder pain also experience rotator cuff pathologies, which can be detected via ultrasound. Please include this point and reference the following article: https://pubmed.ncbi.nlm.nih.gov/37615388/

The hypothesis of the study should be stated at the end of the introduction.

I suggest moving the general information on the stroke participants to the beginning of the results section.

Please include a flow diagram outlining the research process.

In the discussion section, it would be preferable to begin with a summary of the current study in the first paragraph.

·

Basic reporting

no comment

Experimental design

no comment

Validity of the findings

no comment

Additional comments

my question already answered

Reviewer 5 ·

Basic reporting

N/A

Experimental design

N/A

Validity of the findings

N/A

Additional comments

The authors have not addressed my questions, and there is no improvement in this revised version.

---

## Round 0.3 · accepted · Accept

Reviewers’ suggestions have been satisfactorily fulfilled